# Impact of the 1st Wave of the COVID-19 Pandemic and Lockdown on In Utero Transfer Activity in the Paris Area, France

**DOI:** 10.3390/jcm11164850

**Published:** 2022-08-18

**Authors:** Alexandre J. Vivanti, Stanislas Fesquet, Diane Gabriel, Alexandra Letourneau, Catherine Crenn-Hebert, Daniele De Luca, Jean Bouyer, Sophie Novelli, Alexandra Benachi, Raphaël Veil

**Affiliations:** 1Division of Obstetrics and Gynecology, DMU Santé des Femmes et des Nouveau-Nés, Antoine Béclère Hospital, Paris Saclay University, AP-HP, 92140 Clamart, France; 2Groupe de Recherche sur les Infections pendant la Grossesse (GRIG), 75000 Paris, France; 3Epidemiology and Public Health Department, Bicêtre Hospital, Paris Saclay University, AP-HP, 94270 Le Kremlin Bicêtre, France; 4Perinat-ARS-IDF, Regional Health Agency of Ile-de-France (ARS-IDF), 93200 Saint-Denis, France; 5Division of Obstetrics and Gynecology, Louis Mourier Hospital, AP-HP, 92025 Colombes, France; 6Division of Pediatrics and Neonatal Critical Care, DMU Santé des Femmes et des Nouveau-Nés, Antoine Béclère Hospital, Paris Saclay University, AP-HP, 92140 Clamart, France; 7CESP, Inserm, UVSQ, Université Paris-Saclay, 94807 Villejuif, France

**Keywords:** COVID-19, in utero transfer, preterm delivery, lockdown

## Abstract

Background: To assess changes in the number and profile of in utero transfer requests during the first lockdown. Methods: An observational, retrospective, cohort study. All pregnant women, from the Paris area (France), for whom a request for in utero transfer to the transfer unit was made during the first lockdown in France (from 17 March to 10 May 2020) or during a mirror period (years 2016 to 2019) were included. We compared the numbers and proportions of various indications for in utero transfer, the rates of in utero transfer acceptance and the proportion of outborn deliveries. Results: 206 transfer requests were made during the lockdown versus 227, 236, 204 and 228 in 2016, 2017, 2018 and 2019, respectively. The relative proportion of requests for threatened preterm births and for fetal growth restriction decreased from 45% in the mirror period to 37% and from 8 to 3%, respectively. The transfer acceptance rates and outborn deliveries did not differ between time periods. Conclusions: Although a reduction in in utero transfer requests was observed for certain indications, the first lockdown was not associated with a decrease in acceptance rates nor in an increase in outborn births of pregnancies with a high risk of prematurity in the Paris area.

## 1. Introduction

The SARS-CoV-19 epidemic initially broke out in Wuhan, China in late 2019 and then spread globally in the following months. As of 16 March 2020, in France, the authorities reported more than 6600 confirmed cases resulting in the death of 150 patients. On the evening of 16 March, the President of the French Republic decreed a national lockdown starting the following day [1]. The primary objective of the lockdown was to reduce pressure on hospitals, and emergency and intensive care facilities. The announcement of lockdown was followed by a significant drop in primary care consultations, as shown by French health insurance data [2]. The French National Authority for Health quickly understood the existence of major issues concerning the maintenance of continuity of care for pregnant women and their newborns and therefore rapidly issued recommendations [3,4].

During the first lockdown, gynecological emergency units in Ile-de-France hospitals also experienced drastic changes in attendance: a decrease in the number of consultations of more than 40% [5]. The same study also noted a reduction in the number of gynecological hospitalizations in the order of 20%. Regarding perinatal outcomes, the team of Roy Philip was the first to show that lockdown could have an impact on perinatal outcomes: the number of low-birth-weight babies dropped by almost 75% during the first lockdown in Ireland, suggesting a decrease in preterm births [6]. The authors of this study suggested that socioenvironmental and behavioral changes during lockdown could be responsible for this reduction in prematurity (reduced stress, reduced workload, reduced air pollution, etc.). In France, the impact of the COVID-19 pandemic on prematurity has yet to be assessed.

The Paris region was one of the most heavily impacted by the first pandemic wave. The perinatal health system was under great pressure due to the number of patients with COVID-19 requiring adapted maternal and neonatal care [7]. The perinatal health system in France is organized into maternity units of different levels, allowing the intake of newborns according to their weight and gestational age at birth. As inappropriate maternity unit levels for birth increases neonatal morbidity, many regions in France, including Paris, centralize requests for in utero transfers between different maternity unit levels through a dedicated call center for perinatal professionals: the in utero transfer unit. This unit was also put in charge of transferring pregnant women with COVID-19 who required both specific maternal care (the intensive care unit) and perinatal care adapted to gestational age at the time of in utero transfer due to the risk of prematurity.

Our research question was to assess changes in the number and profile of in utero transfer requests during the first 2020 lockdown in the Paris region and to examine whether the overall impact of the pandemic on the health care system may have been responsible for difficulties in transferring pregnant women to appropriate maternity facilities, thus increasing the risk of outborn delivery.

Our primary objective was thus to compare the indications for in utero transfer requests between the 2020 lockdown and a mirror period in previous years. The secondary objectives were to compare between those two periods:-the proportion of successful in utero transfers, overall and by indication for in utero transfer requests;-the proportion of outborn deliveries.

## 2. Materials and Methods

### 2.1. Setting

The perinatal health system in France is organized into four levels of maternity hospitals:-Type III maternity hospitals have a neonatal intensive care unit and are able to receive newborns as soon as the viability threshold is reached.-Type IIA maternity hospitals have a neonatal unit and are able to care for newborns from 32 weeks of age with a birth weight of at least 1500 g.-Type IIB maternity hospitals are similar to type IIA, with an intensive care facility.-Type I maternity hospitals can accommodate healthy newborns from 37 weeks onwards.

An outborn delivery is defined as birth in a maternity hospital that is not appropriate for the gestational age and/or birth weight, with postpartum transfer of the neonate to a tertiary care medical center. If a birth occurs in a maternity unit of an appropriate level, but the newborn must be transferred to a neonatal intensive care unit in another hospital due to lack of available beds, this situation is also considered an outborn delivery. As outborn delivery increases neonatal morbidity [8,9], many regions in France, including the Paris region, centralize requests for in utero transfer through a dedicated call center: the in utero transfer unit. A midwife is available 24/7 to oversee calls from maternity units wishing to transfer pregnant women when there is a risk of spontaneous or induced preterm delivery. The midwife is responsible for finding a place in a suitable maternity unit.

In the Paris region, an average of 178,816 (ranging from 182,684 in 2016 to 172,603 in 2020) annual live births after 22 weeks occurred between 2016 and 2020 (Table A1).

### 2.2. Study Design

This was an observational, retrospective study comparing two groups of patients from different time periods:-“lockdown period”: All pregnant women, from the Paris region, for whom a request for in utero transfer to the Ile-de-France transfer unit was made during the first lockdown in France (from 17 March to 10 May 2020).-“mirror period”: All pregnant women, from the Paris region, for whom a request for in utero transfer to the transfer unit was made during a mirror period (17 March to 10 May to overcome seasonality in births) for the years 2016, 2017, 2018 and 2019.

Exclusion criteria were transfers before 22 weeks (non-viable) or after 37^6/7^ weeks (full-term) and transfers of patients after delivery.

We analyzed data from the in utero transfer unit in the Paris area. We first compared the numbers and proportions of various indications for in utero transfer requests during the lockdown period in 2020 and during the mirror period in previous years (2016 to 2019). Secondly, we compared the “lockdown period” to the “mirror period” in terms of: (i) the rates of in utero transfer acceptance, overall and by transfer indication; and (ii) the proportion of outborn deliveries.

### 2.3. Data Collection

All clinical data are routine care data prospectively collected at the time of the in utero transfer request. The data were then extracted post hoc using the software Hygie-TIU^®^ (SESAN^®^, Paris, France) and implemented in an encrypted Excel^®^ datasheet (Microsoft^®^, Redmond, WA, USA). This included clinical characteristics of the patients and their ongoing pregnancy (maternal age, parity, number of fetuses, gestational age at in utero transfer request, medical history and ongoing treatments), indication for transfer request, type of maternity unit requesting the transfer, outcome of the transfer request and the type of maternity unit in the receiving hospital. Raw data supporting the findings of this study are available from the corresponding author upon reasonable request.

### 2.4. Statistical Analysis

We first compared indications for in utero transfer requests and patient characteristics between the “lockdown period” and “mirror periods”. Quantitative variables were expressed as mean with standard deviation (SD) and were compared using Student’s t test or the Wilcoxon test for variables not normally distributed. Categorical variables were expressed as frequencies and percentages and compared using the chi-square or Fisher exact test. We used the Cochran–Armitage test for the trend for ordinal variables. This description was first performed for all transfer requests regardless of the indication. Then we excluded women whose indication for transfer was severe respiratory symptoms associated with COVID-19. In doing so, we hypothesized that these women represented additional work for maternity hospitals and would not have needed a transfer in the absence of the COVID-19 pandemic. Therefore, their exclusion allowed us to disentangle and isolate the impact of the lockdown itself on transfer requests from the direct effect of the COVID-19 pandemic.

Secondly, we compared the number of transfer requests overall and by indication across years using Poisson regression. We also assessed differences in the proportion of indications for in utero transfer requests between periods using polytomous logistic regression.

Thirdly, we investigated factors associated with effective transfer using logistic regression. Logistic regression models were adjusted for the main potential cofounders related to patient characteristics and maternity settings. Variables with a *p* value less than 0.20 in univariate analyses as well as variables known to be associated with transfer request indications for the first model or transfer acceptance for the second one were included in the final multivariate analysis. For factors associated with transfer acceptance, we also adjusted comparisons for tocolysis and cervical cerclage during pregnancy in a separate additional analysis. This further adjustment should, however, be considered with caution as the exact dates of the administration of these treatments were unknown. One cannot exclude that they may sometimes have been administered following transfer acceptance (or in anticipation of it), leading to reverse causality. Given the low percentage of missing data (1.8% of patients had at least one missing variable for the assessed covariates), models were run on the complete cases dataset.

All statistical tests were two-tailed, and *p* values of less than 0.05 were considered as statistically significant. Analyses were performed using R software version 4.0.5 (R Core Team 2020, Vienna, Austria).

### 2.5. Ethical Approval

This study was approved by the institutional review board of the French College of Obstetricians and Gynecologists (CEROG 2020-OBST-1202) on 15 March 2021. Patient consent was waived due to the noninterventional retrospective study design. All data were de-identified to ensure patient privacy and confidentiality.

## 3. Results

### 3.1. Overall Number of Transfer Requests and Study Population

Over the course of the study, the in utero transfer unit recorded requests for 206 patients during the “lockdown period” (March–May 2020) versus 895 overall in the “mirror periods” of the previous years (227, 236, 204 and 228 in 2016, 2017, 2018 and 2019, respectively). When excluding the 16 women whose indication for transfer request was severe respiratory symptoms associated with COVID-19 in 2020, the number of transfer requests during lockdown decreased to 190. The Poisson regression showed no difference between the “lockdown period” and “mirror periods” in the overall numbers of transfer requests (Table A2, model 1). When excluding from the analysis the patients whose indication for transfer request was severe respiratory symptoms associated with COVID-19 in 2020, we observe a decrease in the number of transfer requests during the lockdown period, which is statistically significant or close to significance when compared to 2016, 2017 and 2019 (Table A2, model 2). These results should be interpreted considering the actual numbers of annual births in Ile-de-France which show a decreasing trend but no major difference between years (Table A1).

The characteristics of the patients for both the lockdown and mirror periods are shown in Table 1. Women were comparable for all maternal and obstetric characteristics between periods (2020 vs. 2016–2019) except for an older maternal age in the “lockdown period” (mean ± SD, 32.1 ± 5.6 versus 30.9 years ± 5.8; *p* = 0.002). There was no significant difference in gestational age when in utero transfer requests were made. Similarly, the types of maternity hospitals requesting in utero transfers did not differ between periods.

### 3.2. Changes in the Indications for In Utero Transfer Request

The numbers of transfer requests per indication and per year are shown in Figure 1. The number of requests for preterm premature rupture of membranes (PPROM), metrorrhagia, pre-eclampsia, hypertension and HELLP syndrome was stable between periods, whereas the number of transfer requests for threatened preterm delivery was lower during the “lockdown period” (*n* = 76) compared to “mirror periods” (*n* varying from 89 to 111). This difference was statistically significant compared to mirror periods in 2017 and 2019 (Poisson model *p* = 0.02 and 0.01, respectively; Table A3). Hence, the proportion of requests for threatened preterm births decreased from 45% during the “mirror period” to 37% during the “lockdown period” (Table 1). Similarly, the number of transfer requests for fetal growth restriction was lower during the “lockdown period” (*n* = 7) compared to “mirror periods” (*n* varying from 12 to 21). The difference was statistically significant compared to 2016, 2017 and 2018 (Poisson model *p* = 0.05, 0.03 and 0.01, respectively; Table A3). Hence, during the “lockdown period”, the proportion of requests for fetal growth restriction decreased to 3% versus 8% in the “mirror period” (Table 1).

We also assessed differences in the proportion of indications for in utero transfer between periods using polytomous logistic regressions. Multivariate models were adjusted for age, multiple pregnancy, history of birth < 1500 g or premature delivery, history of pre-eclampsia and history of late miscarriage or pregnancy with cervical cerclage (Table A4). There were proportionally fewer requests because of the threat of premature delivery and significantly fewer requests because of fetal growth restriction compared to PPROM during the “lockdown period” versus the “mirror period”: OR 0.69 (95% CI 0.46–1.03), *p* = 0.07 and OR 0.37 (95% CI 0.16–0.85), *p* = 0.02, respectively.

### 3.3. Transfer Acceptance

The percentages of transfer acceptance overall and per indication are shown in Table 2. All indications combined, the transfer acceptance rate was 84% (174/206) during the “lockdown period” versus 87% (781/895) in the “mirror period” (Chi2 test, *p* = 0.29). Of the requests for transfer due to symptomatic COVID-19 infection, 15 out of 16 patients were actually transferred. When excluding these 16 women, the percentage of transfer during the “lockdown period” was still 84% (Chi2 test, *p* = 0.19).

In the “mirror period”, rates of transfer acceptance varied from 84% to 90% for PPROM, threatened preterm birth, pregnancy-related vascular complication, metrorrhagia or fetal growth restriction. The rate of transfer acceptance decreased from 69% to 50% for patients whose primary reason for transfer request was labeled as “other reason”. Transfer acceptance rates did not differ between time periods for any indication, except for fetal growth restriction for which it dropped from 57/68 (84%) during the “mirror period” to 4/7 (57%) during the “lockdown period”, although this difference was not statistically significant (Chi2 test, *p* = 0.12).

We then conducted logistic regressions after excluding the 16 women whose indication for transfer request was severe respiratory symptoms associated with COVID-19. The multivariate model was adjusted for the primary reason for transfer, history of birth < 1500 g or premature delivery and the type of maternity hospital requesting the transfer. There was no significant difference in transfer acceptance between years: OR 0.77 (95% CI 0.49–1.25, *p* = 0.30) (Table 3, multivariate model 1). Additional adjustment for tocolysis and cervical cerclage during pregnancy led to similar results (Table 3, multivariate model 2).

### 3.4. Outcomes after Transfer

The rate of outborn deliveries (Table 2) was similar and low for both periods (*n* = 7; 3% during the “lockdown period” versus *n* = 38; 4% during the “mirror period”; Chi2 test, *p* = 0.58). There were no differences in the types of maternity units that received in utero transfers over the two study periods. The indications that motivated the requests for in utero transfer did not influence the likelihood that the transfer would be carried out (Table 3).

## 4. Discussion

This study shows no impact of the first lockdown implemented to fight the SARS-CoV-2 pandemic on the management of in utero transfers in the Paris area. The stakes related to the first pandemic wave were very high due to the significant pressure on the health system. Requests for in utero transfers due to symptomatic COVID-19 infections involved less than 10% of all transfer requests over the same period. There was a great fear that type III maternity units would be overwhelmed, but this study shows no reduction in transfers for conventional obstetric indications and fortunately, no increase in the rate of outborn deliveries.

Interestingly, the number of requests for in utero transfers because of threatened preterm birth and fetal growth restriction decreased during lockdown. Many authors have reported a significant impact of the pandemic, and more specifically, of the different lockdowns on preterm delivery rates and on low-birth-weight births [6,10,11,12,13,14,15,16,17,18,19,20,21,22,23,24,25,26,27,28,29]. Most studies show a decrease in the number of preterm births during successive lockdowns. Lockdown seems to have resulted in a reduction in extreme [18,19] or moderate prematurity [16,25]. The pathophysiological mechanisms behind these changes are not yet known. Several authors have observed a decrease in prematurity in relation to a reduction in induced births without a change in the prevalence of spontaneous prematurity [15,16,22,24]. This suggests the possibility of a reduction in the prevalence of obstetric conditions likely to induce preterm birth (pre-eclampsia and fetal growth restriction). The lockdowns were also periods during which there was a drastic reduction in the number of consultations. The rapid development of tools for remote consultation has helped to limit the impact of the decrease in face-to-face consultations, but it is possible that this situation has contributed to a delay in screening for certain diseases [30]. So, there may have been a lack of screening for certain obstetric conditions, which could partly explain the significant decrease in the number of requests for in utero transfer because of fetal growth restriction.

The COVID-19 pandemic during the first lockdown did not appear to be associated with major difficulties for the effective completion of in utero transfers, despite the existence of transfer requests due to symptomatic COVID-19 infections, as evidenced by the proportion of outborn births which remained very low (3%). This underlines the resilience of the perinatal management system for preterm births in the Paris region, while hospitals in other regions were under great pressure [31].

The first lockdown notably had significant effects on the activity of gynecology–obstetrics departments. Maternity wards experienced a significant decrease in the number of emergency and scheduled consultations. Several countries have reported this trend, such as Israel [32] (35% fewer emergency consultations than in previous years) and the United Kingdom [33] (54% reduction in the number of scheduled appointments). For the latter country, this reduction is explained by appointments that were sometimes not honored but also by a rapid switch to a wide range of teleconsultation services (telephone or video consultations). During the first lockdown, the gynecological emergency departments of the Ile-de-France hospitals also experienced significant changes in attendance: a reduction in the number of consultations of more than 40% was observed during the first lockdown period [5]. The authors of this study also noted a decrease in the number of gynecological hospitalizations of around 20%. Similarly, two Israeli studies evaluated the rates of tubal rupture linked to an ectopic pregnancy: during lockdown, this rate increased two- to three-fold, resulting in an increase in cases of hemoperitoneum in ectopic pregnancies [34,35]. These observations confirm the delay in management induced by a delay in patient consultation.

Our study has several strengths. Firstly, it is a comprehensive analysis involving all in utero transfer activity that transited through the centralized in utero transfer unit in the region of Paris. The study also has very few missing data since it relies on prospectively collected routine care data. The choice of model (the comparison with mirror periods) avoids between-season fluctuations in births. Finally, by focusing on the Paris region, we account for almost 25% of births in metropolitan France [36]. There are, however, certain limitations that should be noted. Firstly, as it was an observational study, any causal interpretation of its results is subject to confounders. Secondly, due to the study design, it is inherently difficult to disentangle the respective effects of the lockdown, the overall context and the disease itself. Still, by excluding the patients whose transfer request indication was symptomatic COVID-19, we managed to isolate the effect of the disease itself, assuming that these women would not have required a transfer had it not been for the infection. It is possible that the observed reduction in fetal growth restriction may be partly related to a decrease in its detection and therefore its management due to a possible decrease in the number of consultations over the same period. Furthermore, our study was limited to the lockdown period and does not allow us to study the effects of the pandemic and lockdown in the longer term. Some requests for in utero transfers may not have been made: in general, any situation where there is a risk of outborn delivery is subject to a request for in utero transfer, unless the situation is particularly unstable, and the patient is not suitable for transport. Unfortunately, this cannot be assessed with the study methodology either. However, we can reasonably assume that these are rare situations. Finally, the data regarding pregnancy outcomes are not available within the framework of this study.

## 5. Conclusions

A reduction in in utero transfer requests was observed in the Paris area for certain indications, namely, fetal growth restriction, during the first lockdown. However, this period was not associated with a decrease in the acceptance rate of in utero transfers requests regardless of indication. The rate of outborn births remained low. This underlines the good capacity of perinatal health systems to adapt despite the strong pressure on hospitals caused by the pandemic.

## Figures and Tables

**Figure 1 jcm-11-04850-f001:**
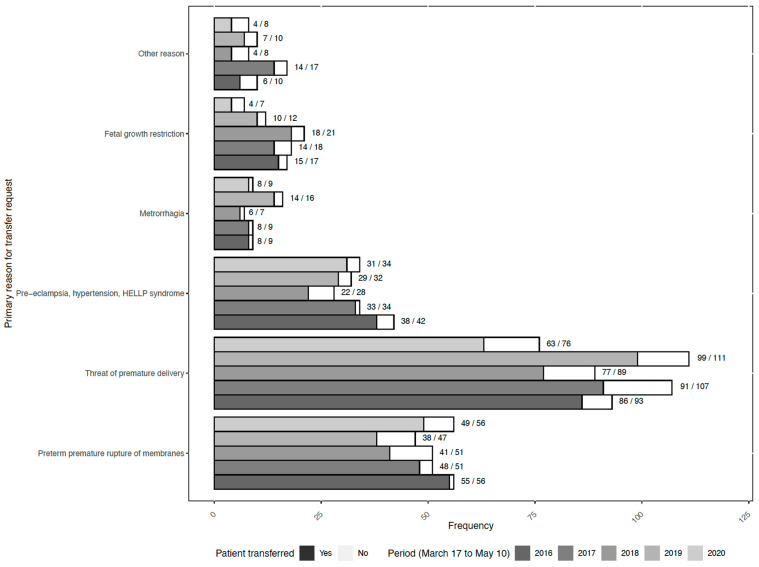
Number of transfer requests per indication during the lockdown period and mirror periods of previous years (2016 to 2019). For each line, the numerator and the denominator represent the number of effective in utero transfers and the number of in utero transfer requests per year and by indication, respectively.

**Table 1 jcm-11-04850-t001:** Characteristics of the patients, their pregnancies and the maternity hospitals that requested in utero transfer according to the study periods.

	“Mirror Period” 17 March–10 May 2016–2019 (*n* = 895)	“Lockdown Period” 17 March–10 May 2020 (*n* = 206)	*p*-Value
Maternal characteristics			
Age (years); mean (SD)	30.9 (5.8)	32.1 (5.6)	0.002
Parity; mean (SD)	1.4 (1.0)	1.5 (1.0)	0.042
ART; *n* (%)	89 (10)	16 (7.8)	0.34
Multiple pregnancy; *n* (%)	112 (13)	31 (15)	0.34
Antihypertensive drug use; *n* (%)	174 (19)	32 (16)	0.19
Tocolysis use; *n* (%)	438 (49)	91 (44)	0.22
Cervical cerclage; *n* (%)	29 (3)	5 (2)	0.54
History of preterm delivery or late miscarriage; *n* (%)	38 (4)	7 (3)	0.58
Gestational age at the time of in utero transfer request			0.72
<24 weeks; *n* (%)	29 (3)	6 (3)	
[24–27^6/7^[; *n* (%)	247 (28)	58 (28)	
[28–31^6/7^[; *n* (%)	389 (43)	82 (40)	
[32–37^6/7^[; *n* (%)	230 (26)	60 (29)	
Indication for in utero transfer request			0.001
PPROM; *n* (%)	205 (23)	56 (27)	
Threatened preterm birth; *n* (%)	400 (45)	76 (37)	
Pregnancy-related vascular complication; *n* (%)	136 (15)	34 (17)	
Metrorrhagia; *n* (%)	41 (5)	9 (4)	
Fetal growth restriction; *n* (%)	68 (8)	7 (3)	
Other reason; *n* (%)	45 (5)	24 (12) *	
Level of maternity unit requesting in utero transfer			0.57
Level I; *n* (%)	164 (18)	36 (18)	
Level IIA; *n* (%)	320 (36)	67 (33)	
Level IIB; *n* (%)	353 (39)	92 (45)	
Level III; *n* (%) ^£^	53 (6)	10 (5)	

£: missing data; *n* = 6. Abbreviations: PPROM: preterm premature rupture of membranes; ART: assisted reproductive technology. * including 16 women (8%) whose primary indication was a severe COVID-19 infection.

**Table 2 jcm-11-04850-t002:** In utero transfer outcomes.

	“Control” Group 17 March–10 May 2016–2019 (*n* = 895)	“Lockdown” Group 17 March–10 May 2020 (*n* = 206)	*p*-Value
Transfer acceptance, overall; *n* (%) ^£^	781 (87)	174 (84)	0.29
Transfer acceptance, by indication; effective transfer/transfer requests (%)			
PPROM	182/205 (89)	49/56 (88)	0.79
Threatened preterm birth	353/400 (88)	63/76 (83)	0.20
Pregnancy-related vascular complication	122/136 (90)	31/34 (91)	>0.99
Metrorrhagia	36/41 (88)	8/9 (89)	>0.99
Fetal growth restriction	57/68 (84)	4/7 (57)	0.12
Severe COVID-19	-	15/16 (94)	-
Other reason	31/45 (69)	4/8 (50)	0.42
Cancellation; *n* (%) ^£^	114 (13)	32 (16)	
Level of maternity hospital receiving in utero transfer			0.51
Level I; *n* (%)	1 (0)	0 (0)	
Level IIA; *n* (%)	37 (5)	10 (6)	
Level IIB; *n* (%)	154 (20)	41 (24)	
Level III; *n* (%)	589 (75)	123 (71)	
Outborn births; *n* (%)	38 (4)	7 (3)	0.58

£: missing data; *n* = 32. Cancellations of transfer requests may be the consequence of a delivery before the actual transfer of the patient or cancellation due to stabilization of the obstetric condition motivating the request.

**Table 3 jcm-11-04850-t003:** Factors associated with transfer acceptance.

	Univariate Model			Multivariate Model 1			Multivariate Model 2		
	OR	95% CI	*p*-Value	OR	95% CI	*p*-Value	OR	95% CI	*p*-Value
Characteristic									
Year, May–March lockdown 2020 vs. mirror period in 2016 to 2019	0.75	(0.49–1.17)	0.19	0.77	(0.49–1.25)	0.27	0.76	(0.48–1.23)	0.25
Primary reason for transfer request									
Preterm premature rupture of membranes	-	-		-	-		-	-	
Threat of premature delivery	0.90	(0.56–1.43)	0.66	0.92	(0.56–1.48)	0.73	0.69	(0.39–1.20)	0.19
Pre-eclampsia, hypertension, HELLP syndrome	1.17	(0.63–2.23)	0.63	1.15	(0.61–2.25)	0.68	1.25	(0.65–2.48)	0.51
Metrorrhagia	0.95	(0.40–2.66)	0.92	0.93	(0.38–2.63)	0.88	0.76	(0.30–2.18)	0.57
Fetal growth restriction	0.57	(0.29–1.16)	0.11	0.52	(0.26–1.09)	0.07	0.58	(0.28–1.22)	0.14
Other reason	0.25	(0.13–0.51)	<0.001	0.27	(0.13–0.56)	<0.001	0.30	(0.14–0.63)	0.001
History of birth weight < 1500 g or preterm delivery	1.63	(0.82–3.73)	0.20	1.67	(0.82–3.85)	0.19	1.77	(0.86–4.14)	0.15
Type of requesting maternity hospital									
I	-	-		-	-		-	-	
IIA	0.86	(0.48–1.48)	0.59	0.88	(0.49–1.54)	0.67	0.86	(0.48–1.51)	0.62
IIB	0.67	(0.38–1.12)	0.14	0.72	(0.41–1.23)	0.24	0.71	(0.40–1.21)	0.22
III	0.27	(0.13–0.56)	<0.001	0.26	(0.12–0.53)	<0.001	0.27	(0.13–0.55)	<0.001
Tocolysis	1.60	(1.12–2.30)	0.01				1.68	(0.99–2.83)	0.05
Cervical cerclage during the current pregnancy	0.48	(0.22–1.16)	0.08				0.43	(0.19–1.07)	0.05

Estimates using logistic regression. The 16 women whose indication for transfer request was severe respiratory symptoms associated with COVID-19 in 2020 were removed from this analysis. Fifteen out of sixteen of them were transferred.

## Data Availability

The data presented in this study are available on request to the corresponding author.

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
