# Peer review of "Impact of the 1st Wave of the COVID-19 Pandemic and Lockdown on In Utero Transfer Activity in the Paris Area, France"

_jcm, 2022, doi:10.3390/jcm11164850_

Round 1

Reviewer 1 Report

Thanks for looking at this - it has certainly been an interesting period in healthcare. Very interesting that the lockdown was so short. 

My main suggestions for improvement given that your findings are limited is to shorten the whole document and distill to the main findings into a short report. I know you have done a lot of work but given the lack of statistical significance and the number of COVID related articles out there, short reports are more relevant and relatable to a COVID fatigued readership. 

I would also recommend deleting the use of the word retardation. In 2022 we say fetal growth restriction as you have said once in the document/tables. 

Author Response

Reviewer 1 – Review Report

Thanks for looking at this - it has certainly been an interesting period in healthcare. Very interesting that the lockdown was so short.

My main suggestions for improvement given that your findings are limited is to shorten the whole document and distill to the main findings into a short report. I know you have done a lot of work but given the lack of statistical significance and the number of COVID related articles out there, short reports are more relevant and relatable to a COVID fatigued readership.

Dear Reviewer,

we take note of your suggestion to shorten the manuscript as a short report, however, we respectfully believe that this manuscript should remain at its present length, which we have already drastically reduced. We would like to leave this manuscript as original research for several reasons:

  • Many clinicians believe that the period of lockdown represented a risk to patients and their fetuses because of possible overload of the health care system. We show in this study that this was not the case for in utero This shows that the health system, in the context of this analysis, was able to overcome the initial difficulties induced by the pandemic.
  • The statistical methodology is specific and involves several models that need to be properly explained.
  • We did not mention this information, but we are the first team to report on the management of prematurity risk situations during the 1st phase of the COVID-19 pandemic.
  • Contrary to what you seem to suggest, a negative result deserves to be disseminated in the same way as a statistically significant result when it can have a clinical impact. Otherwise, it could create a major bias for readers.
  • As you may have noticed from reading this manuscript, we have taken the liberty of including only 3 tables in the body of the manuscript. We have also chosen to include 4 tables in the appendix section. This leaves the choice to the reader to refer to them or not.
  • The instructions for authors of the Journal of Clinical Medicine state: " JCM has no restrictions on the length of manuscripts, provided that the text is concise and comprehensive. Full experimental details must be provided so that the results can be reproduced. "

Therefore, we would like to maintain the length of the manuscript, if not explicitly requested by the editor of JCM.

I would also recommend deleting the use of the word retardation. In 2022 we say fetal growth restriction as you have said once in the document/tables.

This has been modified throughout the manuscript.

Reviewer 2 Report

This is a well written manuscript describing a well conducted, interesting study about the impact of the initial phase of the Covid 19 pandemic on the rate of in utero transfers in the Paris area.

The study shows that the services in Paris were not overwhelmed by the pandemic, which had been a legitimate concern at the start of the pandemic.  The rate of in utero transfer requests was similar to the rate anticipated based on control data from previous years.  When requests based on maternal covid 19 infection were excluded, there was actually a small reduction in the number of request, with specific reductions in requests because of threatened preterm delivery and growth restriction.  This observation is consistent with the findings of other studies that have reported on the impact of the pandemic on rates of prematurity and low birth weight.  The rate of transfer request acceptance remained high.

Some general comments about the study design:

Have the authors considered the impact on the rate of preterm pregnancy loss during this period?  Were some requests not made?

Is this a geographically contained area?  I assume that there is some transfer activity into and out of this region from surrounding areas?  Should this activity be included in the study? 

Some specific comments for consideration by the authors:

Line 95 states “An outborn delivery is defined as birth in a maternity hospital that is not appropriate for the gestational age and/or birth weight, with postpartum transfer of the neonate to a tertiary care medical center”.  In most maternity systems “outborn deliveries” also include those deliveries in another appropriate centre than the one initially intended due to capacity issues in the initially intended delivery centre (ie this could result fomr an in-utero transfer fomr a level 3 centre to another level 3 centre due to capacity issues, but the delivery would still occur in an appropriate centre) .  Was this the case in this study?  If so, it would be helpful to reflect this in the definition on line 95.  If not, it may be better to consider using a term such as “Delivery in an inappropriate centre” rather than “outborn delivery”.

Line 115 – Requests for transfer at full term are excluded from the analysis.  Why is this?  Surely centralisation of care exists for high risk term pregnancies such as those with antenatally recognised malformation?

Figure 1- I don’t understand.  All of the outcomes in each year are expressed as a proportion (eg “Threat of preterm delivery” in 2020 is shown as 63/73).  There is no explanation of what the denominator or numerator mean.  I assume, based on the text, that there were 73 pregnancies with threat of preterm labour in 2020 and that 63 of these were accepted. This should be made clearer in the figure legend, and in the legend to Table 2.

Line 65 “term of birth” – should be “gestation at birth”

Line 235 – “The rate of transfer acceptance decreased to 69% for patients whose primary reason for transfer request was labeled as “other reason”.”  Is this a typographic error?  Looking at Table 2 – the rate falls from 69% in the Mirror period to 50% in the Lockdown period. 

Author Response

Reviewer 2 – Review Report

This is a well written manuscript describing a well conducted, interesting study about the impact of the initial phase of the Covid 19 pandemic on the rate of in utero transfers in the Paris area.

The study shows that the services in Paris were not overwhelmed by the pandemic, which had been a legitimate concern at the start of the pandemic.  The rate of in utero transfer requests was similar to the rate anticipated based on control data from previous years.  When requests based on maternal covid 19 infection were excluded, there was actually a small reduction in the number of request, with specific reductions in requests because of threatened preterm delivery and growth restriction.  This observation is consistent with the findings of other studies that have reported on the impact of the pandemic on rates of prematurity and low birth weight.  The rate of transfer request acceptance remained high.

Thank you for proofreading our manuscript and for your analysis of it, which is consistent with the key messages we wanted to report.

Some general comments about the study design:

Have the authors considered the impact on the rate of preterm pregnancy loss during this period?  Were some requests not made?

No, the methodology of the study did not allow us to assess this data. Only ongoing pregnancies with a suspected prematurity situation were requested to be transferred in utero. Data on the impact of the pandemic on the risk of stillbirth or late miscarriage have been published by other teams.

In general, any situation where there is a risk of outborn delivery is subject to a request for in utero transfer, unless the situation is particularly unstable, and the patient is not suitable for transport. Unfortunately, this cannot be assessed with the study methodology either. However, we can reasonably assume that these are rare situations.

We have added a clarification to the discussion section.

Is this a geographically contained area?  I assume that there is some transfer activity into and out of this region from surrounding areas?  Should this activity be included in the study?

Yes, this is a study carried out within the Ile-de-France in utero transfer unit, which is the region with the highest number of births in France. Transfers outside the region are very limited. Moreover, reporting this information does not appear relevant because in all cases the transfers are carried out in perinatal centers of an appropriate level.

Some specific comments for consideration by the authors:

Line 95 states “An outborn delivery is defined as birth in a maternity hospital that is not appropriate for the gestational age and/or birth weight, with postpartum transfer of the neonate to a tertiary care medical center”.  In most maternity systems “outborn deliveries” also include those deliveries in another appropriate centre than the one initially intended due to capacity issues in the initially intended delivery centre (ie this could result fomr an in-utero transfer fomr a level 3 centre to another level 3 centre due to capacity issues, but the delivery would still occur in an appropriate centre) .  Was this the case in this study?  If so, it would be helpful to reflect this in the definition on line 95.  If not, it may be better to consider using a term such as “Delivery in an inappropriate centre” rather than “outborn delivery”.

Your comment is very relevant and deserves clarification in the definition we make in the manuscript regarding the levels of centers. If a birth takes place prematurely in a level III center, but the neonate is subsequently transferred to a neonatal intensive care unit in another hospital, the birth is considered an outborn delivery. You can see in table 3 that 5 to 6% of the requests for in utero transfers come from level III centers (a typographical error crept into the table and has been corrected in the revised version).

Level of maternity unit requesting in utero transfer

Level I; n (%)

164 (18)

36 (18)

Level IIA; n (%)

320 (36)

67 (33)

Level IIB; n (%)

353 (39)

92 (45)

P-value 0.57

Level III; n (%)£

53 (6)

10 (5)

We have added a clarification to the materials and methods section: “If a birth occurs in a maternity unit of appropriate level, but the newborn must be transferred to a neonatal intensive care unit in another hospital due to lack of available beds, this situation is also considered an outborn delivery.”

Line 115 – Requests for transfer at full term are excluded from the analysis.  Why is this?  Surely centralisation of care exists for high risk term pregnancies such as those with antenatally recognised malformation?

Transfer requests for full-term patients are not managed by the in utero transfer unit. It is therefore not possible for us to have this information. However, this does not seem to us to be a limitation of this study, as such requests are made in the vast majority of cases in the event of temporary saturation of the requesting maternity ward.

Figure 1- I don’t understand.  All of the outcomes in each year are expressed as a proportion (eg “Threat of preterm delivery” in 2020 is shown as 63/73).  There is no explanation of what the denominator or numerator mean.  I assume, based on the text, that there were 73 pregnancies with threat of preterm labour in 2020 and that 63 of these were accepted. This should be made clearer in the figure legend, and in the legend to Table 2.

We have added a clarification: “For each line, the numerator and the denominator represent the number of effective in utero transfers and the number of in utero transfer requests per year and by indication, respectively.”

Line 65 “term of birth” – should be “gestation at birth”

This has been corrected.

Line 235 – “The rate of transfer acceptance decreased to 69% for patients whose primary reason for transfer request was labeled as “other reason”.”  Is this a typographic error?  Looking at Table 2 – the rate falls from 69% in the Mirror period to 50% in the Lockdown period.

The correction has been made: “The rate of transfer acceptance decreased from 69% to 50% for patients whose primary reason for transfer request was labeled as “other reason”.”
